

# Analysis of ergot alkaloid gene expression and ergine levels in different parts of *Ipomoea asarifolia*

Yanisa Olaranont, Alyssa B. Stewart, Wisuwat Songnuan and Paweena Traiperm

Department of Plant Science, Faculty of Science, Mahidol University, Bangkok, Thailand

## ABSTRACT

**Background:** Ergot alkaloids are renowned for their pharmacological significance and were historically attributed to fungal symbioses with cereal crops and grasses. Recent research uncovered a symbiotic relationship between the fungus *Periglandula ipomoea* and *Ipomoea asarifolia* (Convolvulaceae), revealing a new source for ergot alkaloid synthesis. While past studies have emphasized the storage of both the fungus and alkaloids in leaves and seeds, recent work has found they also occur in other plant parts. This study aimed to examine expression of the *dmaW* gene, which plays a crucial role in ergot alkaloid biosynthesis, and to quantify ergot alkaloid levels across various organs and growth stages of *I. asarifolia*.

**Results:** Our findings revealed the highest levels of *dmaW* gene expression in young seeds and young leaves, whereas the highest ergine concentrations were found in mature leaves followed by young leaves. In light of previous studies, we propose three hypotheses to reconcile these conflicting results: the possibility of an inefficient ergot alkaloid biosynthesis pathway, the possibility that different types of ergot alkaloids are produced, and the existence of an ergot alkaloid translocation system within the plant. Furthermore, ergine concentration and ergot alkaloid biosynthesis gene expression were detected in stems, roots, and flowers, indicating that ergot alkaloids are produced and accumulated in all studied parts of *I. asarifolia*, rather than being solely confined to the leaves and seeds, as previously reported.

**Conclusions:** Overall, our study reveals that ergot alkaloids are produced and accumulated in most parts of *I. asarifolia*, suggesting a plant-wide biosynthesis and potential transport system, challenging the previous belief that biosynthesis was confined to glandular trichomes on leaves.

## INTRODUCTION

Ergot alkaloids are mycotoxins produced by several species of fungi, particularly those belonging to the Clavicipitaceae family. One well-known representative is the *Claviceps purpurea* fungus, which infects a wide range of cereal crops and grasses (*e.g.*, rye, wheat, barley, oats, millet, and triticale) and replaces the host grain with dark, ergot-filled structures called sclerotia or ergots (*Flieger, Wurst & Shelby, 1997*; *Krska & Crews, 2008*; *Babič et al., 2020*; *Berraies et al., 2024*). Interest in ergot alkaloids grew after the discovery

Corresponding author
Paweena Traiperm,
paweena.tra@mahidol.edu

of their pharmacological properties and clinical applications (*Chen et al., 2017*). For instance, ergometrine is widely used in postpartum treatment as it stimulates uterine contraction (*De Costa, 2002*). Ergotamine and its derivatives show remarkable anti-migraine efficacy through their interactions with receptors in the central nervous system (*Silberstein & McCrory, 2003*; *Saper & Silberstein, 2006*). Additionally, several ergot alkaloids, such as α-dihydroergocryptine, bromocriptine, cabergoline, and pergolide, are effective in treating Parkinson's disease (*Bergamasco et al., 2000*; *Mizuno et al., 2003*; *Curran & Perry, 2004*; *Van Camp et al., 2004*). Cabergoline is also commonly used as a first-line agent in the treatment of prolactinomas (*Corsello et al., 2003*; *Ono et al., 2008*; *Molitch, 2014*; *Shimon et al., 2016*).

In addition to cereal crops and grasses, some plant species in the Convolvulaceae family have been reported to contain toxic bioactive compounds, including ergot alkaloids, indole-diterpene alkaloids, and swainsonine (*Lee, Gardner & Cook, 2017*; *Cook et al., 2019*; *Beaulieu et al., 2021*). Previous research has shown that these ergot alkaloids in Convolvulaceae species are produced by a new genus of symbiotic fungi in the Clavicipitaceae family, *Periglandula* (*Kucht et al., 2004*; *Steiner et al., 2011*; *Steiner & Leistner, 2012*). These fungi contain the dimethylallyl-tryptophan synthase gene (*dmaW*), which is the determinant step in ergot alkaloid biosynthesis (*Steiner et al., 2011*). The types of ergot alkaloids vary among Convolvulaceous host plant, including chanoclavine, lysergic acid, lysergol, elymoclavine, agroclavine, lysergic acid amide (ergine), lysergic acid α-hydroxyethylamide (LAH), ergonovine (ergometrine), penniclavine, setoclavine, festuclavine, ergobalansine, ergonovine, cycloclavine, and others (*Beaulieu et al., 2015*; *Nowak et al., 2016*; *Steiner & Leistner, 2018*). *Ipomoea asarifolia* (Desr.) Roem. & Schult. has been found to contain chanoclavine, ergine, LAH, ergonovine, and ergobalansine (*Ahimsa-Müller et al., 2007*; *Steiner & Leistner, 2018*), while other host plant species are known to contain different ergot alkaloids.

*Periglandula* fungi are primarily associated with the leaves and seeds of their host plants (*Ahimsa-Müller et al., 2007*; *Steiner & Leistner, 2018*), and ergot alkaloids have been detected in both the leaves and seeds of the plants (*Markert et al., 2008*; *Nowak et al., 2016*). Our recent study examined numerous parts of *Ipomoea asarifolia* and discovered *Periglandula ipomoeae* on six out of eight plant parts: young folded leaves, mature leaves, flower buds, mature flowers, young seeds, and mature seeds (*Olaranont et al., 2022*). However, it is still uncertain which parts of *I. asarifolia* (both fungus-associated and non-associated) contain ergot alkaloids. To confirm the sites of ergot alkaloid biosynthesis in *I. asarifolia*, this study examined gene expression levels of the *dmaW* gene, which is essential for the determining step in ergot alkaloid biosynthesis (*Steiner et al., 2006*; *Ahimsa-Müller et al., 2007*), and quantified the amount of ergot alkaloids in different parts of *I. asarifolia*. Such knowledge will better our understanding of symbioses between ergot alkaloid-producing fungi and their host plants. Portions of this text were previously published as part of a preprint (*Olaranont et al., 2024*).

## MATERIALS AND METHODS

### Sample collection

We collected *Ipomoea asarifolia* (Desr.) Roem. & Schult. from its natural habitat in Na Di District, Prachin Buri province, Thailand (14°08′N, 101°44′E). The plant was identified by P. Traiperm and voucher specimen (Y. Olaranont 07) is stored in the Department of National Parks, Wildlife and Plant Conservation, Bangkok, Thailand. All institutional and national guidelines and legislation were adhered to in the production of this study. Permission to collect plant specimens was granted by private land owners (Mr. Sorraton Sookmark). We collected the following plant parts and developmental stages of interest: young (folded) leaves, mature (opened) leaves, stems, roots, flower buds, mature flowers, young (still green) seeds, and mature (completely ripe) seeds. We kept samples on ice during transport to the lab and, upon arrival, transferred gene expression samples to a $-80\,°C$ freezer and placed ergot alkaloid quantification samples in an oven at $40\,°C$ to dry. Moreover, before conducting gene expression and ergot alkaloid analyses, we first wanted to ensure that the fungus was present on our study plants. We therefore used additional fresh young leaves that we had collected from our study plants to confirm the presence of the fungus *via* staining (see below) before proceeding with the remaining analyses.

### Confirmation of fungal presence

We confirmed the presence of the fungus using a staining method with lactophenol cotton blue (*Sangeetha & Thangadurai, 2013*). We cut fresh young leaves and soaked them in 10% KOH until clear. After a gentle rinse with water, we stained the adaxial surface of the samples with lactophenol cotton blue, which stained the fungal hyphae blue and facilitated observation under a light microscope (Olympus CX21). We previously confirmed that the fungal species was *Periglandula ipomoeae* using molecular analysis (*Olaranont et al., 2022*). As all examined leaves were found to exhibit the fugus, we proceeded to use our remaining samples for the gene expression and ergot alkaloid analyses.

### Total RNA extraction and cDNA synthesis

We extracted total RNA from frozen samples (stored at $-80\,°C$) of each plant part (five replicates per plant part) using the CTAB method (slightly modified from *Morante-Carriel et al. (2014)*). We ground each sample with liquid nitrogen and incubated approximately 200 mg of the ground sample with a CTAB extraction buffer (300 mM Tris–HCl, 25 mM EDTA, 2 M NaCl, 2% CTAB, 2% PVPP, and 2% β-mercaptoethanol) at $65\,°C$, before treating samples with chloroform–isoamylalcohol (24:1), 3 M sodium acetate, isopropanol, and 10 M LiCl. We purified total RNA by treating it with DNase using the DNA-free™ DNA Removal Kit (Invitrogen, Waltham, MA, USA). We then used the purified RNA as a template for synthesizing first-strand cDNA with the iScript™ cDNA Synthesis Kit (Bio-Rad, Hercules, CA, USA) following manufacturer's protocol. We stored the synthesized cDNA at $-20\,°C$ until use.

## Quantitative real-time PCR

We used quantitative RT-PCR (qRT-PCR) to measure expression of the *dmaW* gene compared to three reference genes, all of which are fungal genes. Primers for the *dmaW* gene and reference genes (*actG, atp6, tefA*) are listed in Table 1. We developed primers using sequences from the NCBI database. We BLAST searched each primer against the NCBI database and found matches only with *Periglandula*, with no match to the plant sequences from several related plant species, providing a limited level of support for their specificity to the fungus. We performed Sanger sequencing of the amplicons to confirm that the products originated from the gene of interest, and the sequences have been deposited in the NCBI database under accession numbers PV410212–PV410219. We conducted qRT-PCR using KAPA SYBR® FAST qPCR Master Mix (2X) with 10 ng of template per reaction, set for 35 amplification cycles. We analyzed each of the five samples per plant part in triplicate using 7,500/7,500 Fast Real-Time PCR Systems (Applied Biosystems, Foster City, CA, USA). We calculated the relative expression of the *dmaW* gene using the $2^{-\Delta Ct}$ method (*Livak & Schmittgen, 2001*). The Ct value of the *dmaW* gene was compared to the average Ct value of the three reference genes in each plant part to obtain the ΔCt. The Ct value for the no-template control was recorded as 'undetermined'.

## Ergot alkaloid analysis

We quantified ergine concentration (a representative of ergot alkaloids) using high-pressure liquid chromatography (Waters Alliance 2695 HPLC). We dried fresh samples (three replicates per plant part) at 40 °C in forced convection oven for 3 days. The dried samples were stored in a closed container with silica gel before alkaloid extraction. We then ground each sample until it was fine enough to pass through a 425-μm sieve. To prevent cross-contamination, the mortar and pestle were thoroughly washed with chloroform and methanol between each sample. We soaked approximately 50 mg of the fine powder in 1 ml of gradient-grade methanol (Lichrosolv® by Merck) for 3 days at 4 °C and vortexed samples daily (*Beaulieu et al., 2015*). We filtered the methanol extract of each sample through a 0.45-μm filter and analyzed the extracts *via* reverse-phase HPLC on a C18 column (Zorbax Eclipse Plus C18: 4.6 × 150 mm i.d., 5 μm; Agilent, Santa Clara, CA, USA) with a fluorescence detector set to excitation and emission wavelengths of 310 and 410 nm, respectively. We performed the multilinear gradient procedure following *Panaccione et al. (2012)*. We used commercial lysergamide (ergine) as a standard solution (purchased from TRC Canada) and diluted it with acetonitrile to create calibration standards at 50, 500, 1,500, 3,000, and 6,000 ng/ml. We used the Empower Chromatography Data System for quantitative analysis of ergine.

## Statistical analysis

We performed all statistical analyses in R version 4.2.1 (*R Core Team, 2016*). We compared differences in relative gene expression across plant parts using the nonparametric Kruskal-Wallis test with Wilcoxon's multiple comparisons test, given the non-normality of
**Table 1 List of primers used in this study.**

| Gene | Primer sequences | | Product size (bp) | E (%) | $R^2$ | Accession number |
|------|------|------|------|------|------|------|
| | Forward | Reverse | | | | |
| *dmaW* | CACATTAGCGATTCTGGAGA | ATTGGTTTCCCTTGAGATCC | 194 | 92.57 | 0.999 | KP235276 |
| *actG* | AGGACTCATATGTCGGTGAC | TGCCAGATCTTCTCCATATC | 112 | 108.25 | 0.985 | HQ702608 |
| *atp6* | GTTCCTTATAGCTTCGCATC | AGGACATCCAGCAGGTACTA | 141 | 97.85 | 0.997 | HQ702614 |
| *tefA* | GTGAAGAACGTCTCCGTAAA | GTACGTCGGTCAATCTTCTC | 215 | 96.89 | 0.989 | KP689568 |

Note:

*dmaW*, Dimethylallyl tryptophan synthase; *actG*, gamma-actin; *atp6*, ATP synthase subunit A; *tefA*, translation elongation factor 1-alpha; E, PCR efficiency (calculated from the standard curve); $R^2$, regression coefficient.

the residuals (Shapiro–Wilk test). We compared differences in ergine concentrations across plant parts using a one-way analysis of variance (ANOVA) with Tukey's multiple comparisons test. Finally, we examined the correlation between relative gene expression and ergot alkaloid concentration using Pearson's correlation coefficient.

## RESULTS

Fungal presence on *I. asarifolia* was confirmed *via* staining with lactophenol cotton blue. Blue fungal mycelia were detected on the adaxial surface of all young leaves examined (Fig. 1A). Fungal hyphae were observed associating with plant glandular trichomes (Fig. 1B). Our previous experiment confirmed that this fungus is *P. ipomoea* (*Olaranont et al., 2022*).

The highest relative expression of the *dmaW* gene was found in young seeds, followed by young leaves, flower buds, mature leaves, mature seeds, mature flowers, stems, and roots, respectively (Fig. 2). The expression level of the *dmaW* transcript in young seeds was 200 times greater than in roots, which had the lowest expression levels. Comparing young and mature stages revealed higher expression in young leaves, flower buds, and young seeds compared to their mature counterparts (four, three, and 15 times greater expression, respectively). Our results revealed significant differences in relative expression levels across all plant parts ($\chi^2 = 98.154$, $p < 0.01$), except for between mature leaves and flower buds and between mature flowers and mature seeds (Fig. 2). Similar to our *dmaW* results, expression of the three reference genes (*actG*, *atp6*, and *tefA*) was highest in young leaves and young seeds (Fig. S1A). The positive correlation between *dmaW* expression and that of the three reference genes in different plant parts was significant (Fig. S1B; $r = 0.841$, $p = 0.009$).

The concentration of ergot alkaloids in *I. asarifolia* was analyzed and quantified through the use of ergine as a representative. Statistical analysis of ergine concentration revealed significant differences across plant parts (F = 963, $p < 0.01$). The highest concentration of ergine was found in mature leaves (68.7 µg/g), followed by young leaves, young seeds, mature seeds, stems, flower buds, roots, and mature flowers (48.97, 15.24, 11.71, 4.41, 4.16, 3.20, and 1.72 µg/g, respectively) (Fig. 3). Similar to our gene expression results, young plant parts tended to exhibit higher levels of ergine compared to mature parts, as seen in

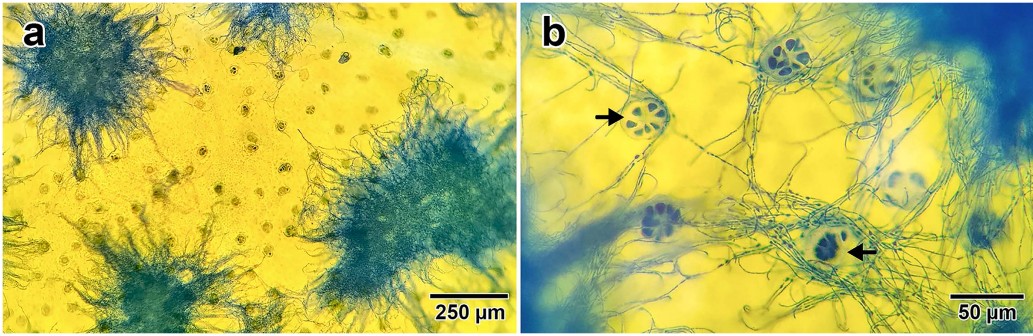

**Figure 1 Fungal hyphae stained by lactophenol cotton blue.** (A) Groups of hyphae on the adaxial surface of a young leaf. (B) Close up photo showing the association between fungal hyphae and glandular trichomes (indicated by black arrows) on the leaf surface.

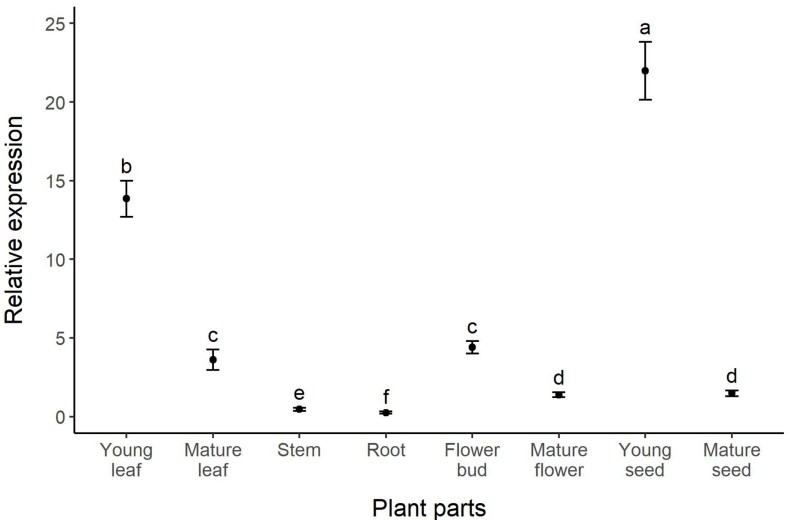

**Figure 2 Relative expression (fold change) of the *dmaW* gene in different plant parts of *I. asarifolia*.** Expression was relative to the reference genes *actG*, *atp6*, and *tefA*. Plant parts with different letters are significantly different ($p < 0.05$).

flowers and seeds. However, mature leaves were found to contain a higher concentration of ergine compared to young leaves. There were no significant differences in concentration among stems, roots, flower buds, and mature flowers, where only minimal amounts of ergine were found (Fig. 3).

The correlation between relative gene expression and ergine concentration was not significant (r = 0.266, $p$ = 0.525; Fig. 4). While some plant parts had comparable levels of gene expression and ergine concentrations, other plant parts exhibited unexpected results. Notably, mature leaves exhibited relatively low gene expression but had the highest concentrations of ergine, while young seeds exhibited the highest levels of gene expression but had only moderate concentrations of ergine.

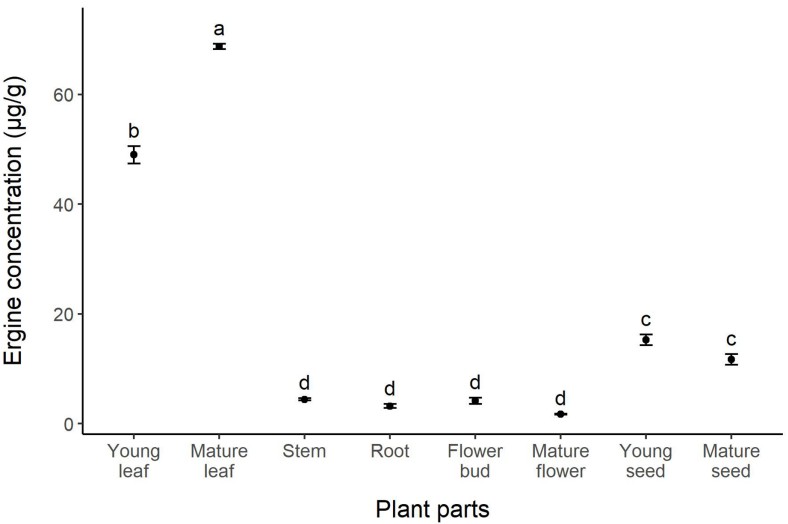

**Figure 3 Concentration of ergine in different parts of *I. asarifolia*.** Plant parts with different letters are significantly different ($p < 0.05$).

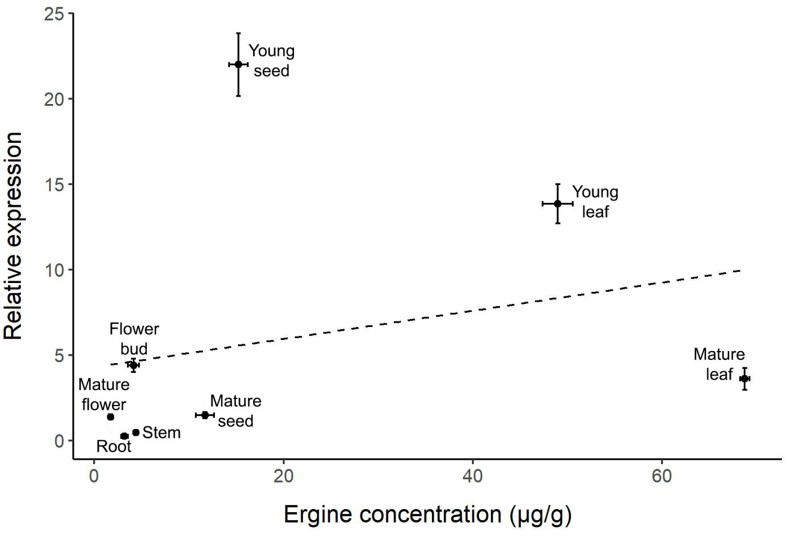

**Figure 4 Correlation between relative *dmaW* expression and ergine concentration for the eight plant parts examined in *I. asarifolia*.** (r = 0.266, p = 0.525).

## DISCUSSION

This study reveals the relative expression levels of the *dmaW* gene, which is required for the determinant step in ergoline alkaloid biosynthesis, in eight different parts of *I. asarifolia*. The expression of the gene was found in all parts of the plant, with the highest levels in young seeds and young leaves, and the lowest in roots. In addition, ergine concentration was also detected in all studied parts, with the highest concentration in mature and young leaves, and the lowest in mature flowers. Our study is the first to develop

primers and investigate *dmaW* expression, as well as the first report of ergot alkaloid quantification across multiple parts of the host plant, not just leaves and seeds.

Differential gene expression of *dmaW* was observed across the studied plant parts of *I. asarifolia*. The three parts of *I. asarifolia* found to have the highest *dmaW* expression levels were young seeds, young leaves, and flower buds. When comparing the expression of the three reference genes to each other, the fold differences across plant organs were similar, which also confirms variation in fungal biomass across different plant parts, with the highest biomass in young seeds and young leaves (Fig. S1A). These results correspond with our previous observation of high densities of fungal hyphae on these same plant parts (*Olaranont et al., 2022*). Nonetheless, the relative expression of the *dmaW* gene was still found to be exceptionally high in both young seeds and young leaves, even when compared with the average reference gene expression level. These results suggest that these two organs do not just harbor substantial fungal biomass, but that the fungi within these organs also have elevated expression of the ergot alkaloid biosynthesis gene (Fig. S1B). Thus, it appears that ergot alkaloid production is notably amplified in young seeds and young leaves.

One possible explanation for differential *dmaW* expression is that genes can have specific functions within different plant tissues. Expression level and site are known to vary based on function and environmental conditions (*Alberts et al., 2002*; *Palande et al., 2022*). Previous research has demonstrated that environmental or abiotic stress can impact the relative expression of genes involved in secondary metabolite biosynthesis in fungi (*Medina et al., 2015*). The fungal genus *Metarhizium*, belonging to the Clavicipitaceae family, has been observed to produce ergot alkaloids under certain conditions but not others (*Leadmon et al., 2020*). Comparing the high level of gene expression in young seeds with the low level in mature seeds observed in our study, it is possible that the soft and moist tissue of young seeds provides a more suitable environment for fungal growth and functioning, compared to the dry conditions experienced in hard mature seeds. Variation in gene expression can also be attributed to the specific functions of the gene. For instance, when *Cucurbit* species were subjected to heat shock, gene expression associated with heat tolerance differed between stems and roots (*Ara et al., 2013*). The *dmaW* gene is known to play a vital role in synthesizing ergot alkaloids. These alkaloids are believed to serve as a defense mechanism for the plant against insects and herbivores (*Jakubczyk, Cheng & O'Connor, 2014*; *Beaulieu et al., 2021*), with the fungi receiving protection and nutrition from the plant in return (*Jakubczyk, Cheng & O'Connor, 2014*). On the other hand, it is also possible that the fungi produce ergot alkaloids for their own benefit, but these compounds also happen to be beneficial to their host plants. In *I. asarifolia*, a possible explanation for high fungal biomass and *dmaW* gene expression in young seeds and young leaves may be due to the higher nutrient content in these organs compared to others. Further insights will be gained once the specific function of the gene is better understood.

Our gene expression analysis also revealed two other noteworthy results. First, we detected low levels of *dmaW* expression in both the stems and roots of *I. asarifolia*. This finding is noteworthy given that previous studies were not able to confirm the presence of

*Periglandula* in the stems and roots of its host plant through molecular, histochemical, or anatomical techniques (*Olaranont et al., 2022*). Thus, it appears that stems and roots do host small quantities of the fungus, which can be detected with qRT-PCR due to its higher sensitivity compared to normal PCR (*Deprez et al., 2002*; *Peters et al., 2004*). Second, we observed higher expression levels in the young stages of the plant parts, as compared to the mature stages, and this also corresponds with the fungal densities reported in our previous study (*Olaranont et al., 2022*). However, while the densities of fungal hyphae in flower buds and mature flowers were similar (*Olaranont et al., 2022*), gene expression levels in flower buds were significantly higher than in mature flowers (Fig. 2). Thus, the higher *dmaW* expression observed in young developmental stages is not solely due to fungal density, and additional research is still needed to understand the drivers of *dmaW* expression.

We also observed differences in the concentrations of ergine (a representative of ergot alkaloids) across different parts of the host plant. Previous research has shown that ergot alkaloids are produced by clavicipitaceous fungi, not the host plants themselves (*Kucht et al., 2004*; *Markert et al., 2008*), and several Convolvulaceae species were previously found to contain ergot alkaloids, but only the leaves and seeds had been studied (*Markert et al., 2008*; *Beaulieu et al., 2013*; *Nowak et al., 2016*; *Beaulieu et al., 2021*). Our findings show that leaves are the major accumulation site of ergine, followed by seeds, with other plant parts containing low concentrations of ergine (Fig. 3). Variation in ergot alkaloid concentration across plant parts was expected. A study on *I. asarifolia* and *Turbina corymbosa* (L.) Raf found that even different developmental stages of leaves contain unequal amounts of ergot alkaloids (*Steiner et al., 2015*). Similarly, *Beaulieu et al. (2013)* reported that in some Convolvulaceae species the total amount of ergot alkaloids differed significantly in different parts of the seed (cotyledon, embryonic axis, endosperm, and seed coat). The authors suggested that the differential allocation of ergot alkaloids may be influenced by the plant's need for protection against pests and diseases, as these alkaloids are known for their defensive qualities (*Beaulieu et al., 2013*). If the seedlings are exposed to more pests and diseases above ground, then it is expected that they will allocate more ergot alkaloids to their cotyledons and hypocotyl. Conversely, if the species experiences higher pressure from below-ground pests like nematodes or soil-borne pathogens, then it is anticipated that more ergot alkaloids will be allocated to their roots (*Beaulieu et al., 2013*). The leaves of *I. asarifolia* were reported to be consumed by various leaf-eating insects, including several species of cassidid and chrysomelid beetles, lepidopterous larvae, and hesperiid larvae (*Carroll, 1978*; *Laurent et al., 2003*; *Ghostin et al., 2007*; *Silva, Medeiros & Kerpel, 2021*). The significantly greater concentrations of ergine found in *I. asarifolia* leaves, compared to stems, roots, flowers, and seeds, therefore suggests that *I. asarifolia* may require the most protection from foliar pests and pathogens. Although there are no reports of seed predators specifically targeting *I. asarifolia*, other species in the *Ipomoea* genus, such as *I. pes-caprae* and *I. nil*, are known to be preyed upon by bruchid beetle larvae (*Devall & Thien, 1989*; *Helman, Sobrero & Rana, 2020*). Such seed predation could explain why the seeds of *I. asarifolia* contain a moderate amount of ergot alkaloids—either

as a defense mechanism against potential seed predators or as a trait inherited through close evolutionary relationships within the *Ipomoea* genus.

Interestingly, we did not observe a correlation between ergot biosynthesis gene expression and ergine concentration across plant parts. Two outliers are particularly noteworthy: young seeds had the highest *dmaW* expression but only moderate ergine concentrations, while mature leaves had low gene expression but the highest concentrations of ergine. The high ergine levels in mature leaves are reasonable, as ergine likely accumulates over time following high *dmaW* expression during earlier developmental stages in young leaves. However, a conflict remains in the young seeds, which had the highest *dmaW* expression but exhibited lower ergine concentration compared to leaves. Several factors may contribute to these results. First, the biosynthesis pathway of ergot alkaloids is inefficient and it is possible that the necessary precursors needed for ergine production in each part of the plant vary in quantity (*Panaccione, 2005*). Therefore, even if gene expression is high, as we observed in young seeds, ergine concentrations may be relatively low if there are not enough precursors. Instead, these plant parts may end up containing other intermediates or end products (*i.e.*, other ergot alkaloids) instead of producing high amounts of ergine. Other ergot-producing fungi (*e.g.*, *Epichloë*, *Claviceps*, and *Aspergillus*) were also found to have inefficient biosynthesis pathways, which may actually be advantageous if they produce diverse alkaloids within the fungus or its host plant (*Beaulieu et al., 2015*; *Florea, Panaccione & Schardl, 2017*).

Second, our study only examined one type of ergot alkaloid (ergine) due to the inability to obtain the other types of ergot alkaloids. Yet *I. asarifolia* is known to contain other ergot alkaloids as well, such as chanoclavine, LAH, ergonovine, and ergobalansine (*Steiner & Leistner, 2018*). Therefore, it is possible that high *dmaW* expression can lead to the biosynthesis of ergot alkaloids other than ergine. The *dmaW* gene initiates the first step of the ergot alkaloid biosynthesis pathway, potentially resulting in a variety of ergot alkaloids beyond just ergine during subsequent steps in the cascade (*Liu, Panaccione & Schardl, 2009*; *Young et al., 2015*). According to the ergot alkaloid biosynthetic pathway summarized by *Florea, Panaccione & Schardl (2017)*, ergine, LAH, ergonovine, and ergobalansine are produced in the later stages of the pathway, whereas chanoclavine is synthesized in the early steps. The regulation of this biosynthetic pathway remains unclear, particularly how plants allocate metabolic flux to favor the production of specific metabolites. Different plant species exhibit distinct patterns of biosynthetic regulation. For instance, in *I. argillicola*, ergobalansine is the most abundant ergot alkaloid, followed by ergonovine, LAH, ergine, and chanoclavine, respectively. In contrast, *I. tricolor* predominantly produces ergine, followed by chanoclavine, LAH, and ergonovine (*Beaulieu et al., 2013*). This diversity in ergot alkaloid composition, concentration, and distribution during morning glory ontogeny suggests a highly regulated process, potentially reflecting evolutionary selection for defense against pests and pathogens (*Beaulieu et al., 2013*).

A third factor that may contribute to the conflicting gene expression and ergine concentration results is differential ergot alkaloid translocation and accumulation within

the plant. If the sites of metabolite biosynthesis and accumulation differ, the correlation between gene expression and compound levels can be relatively low (*Delli-Ponti, Shivhare & Mutwil, 2021*). For example, nicotine alkaloids are synthesized in the root before being transported and accumulated in the aerial parts of the tobacco plant (*Katoh et al., 2005*). The transportation of ergoline alkaloids between and within *Periglandula* and Convolvulaceous plants is complex and still unclear. The first part of the process involves exchanging sesquiterpene and ergot alkaloids between the fungi and host plant, and the second part involves distributing the ergot alkaloids throughout different parts of the plant (*Steiner et al., 2015*; *Leistner & Steiner, 2018*). One study published prior to the discovery of ergot alkaloid-producing fungi reported that ergot alkaloids in the leaves of *I. tricolor* can be translocated throughout the plant (*Mockaitis, Kivilaan & Schulze, 1973*). *Beaulieu et al. (2013)* also reported the differential allocation of ergot alkaloids within Convolvulaceae plant species. Detailed information about how the plant translocates ergot alkaloids is still unknown. Previous studies have reported three families of plant alkaloid transporters, including ATP-binding cassette (ABC) proteins, multidrug and toxic compound extrusion (MATE) transporters, and purine permease (PUP) families (*Yazaki, 2006*; *Shitan, Kato & Shoji, 2014*), which could potentially serve as channels for transporting ergot alkaloids in *I. asarifolia*. In addition to these transporters, there are two other potential mechanisms for alkaloid transport: simple diffusion followed by membrane trapping and vesicle-mediated transport (*Shitan & Yazaki, 2007*; *Shitan, Kato & Shoji, 2014*). The relationship between gene expression and metabolite concentration is complex and further research is required to fully understand ergot alkaloid synthesis, transport, and storage.

## CONCLUSIONS

It was previously a matter of debate as to where the main site of ergot alkaloid biosynthesis was located. According to previous studies, glandular trichomes on the leaves of the plant were considered to be the prerequisite site for ergoline alkaloids, thus, the leaves were believed to be the biosynthesis site of ergot alkaloids, some of which were then transported to the flowers and seeds of the host plant (*Steiner et al., 2015*; *Leistner & Steiner, 2018*). However, our findings indicate that ergot alkaloids are produced and accumulated in most, if not all, parts of *I. asarifolia*. Additionally, there may be a transportation system in place to distribute the compounds throughout the various parts of the plant. Although our study only quantified ergine, a broader analysis using multiple types of ergot alkaloids could offer a clearer understanding of each metabolite's synthesis, transport, and storage. To date, at least 40 species of morning glory have been found to contain ergot alkaloids (*Eich, 2008*; *Beaulieu et al., 2021*), and the seeds of these Convolvulaceae species can contain ergot alkaloids at 1,000 times greater concentrations than the seeds of grasses infected by other clavicipitaceous fungi (*Kucht et al., 2004*; *Ahimsa-Müller et al., 2007*). Considering the beneficial pharmaceutical properties of ergot alkaloids, the biosynthesis gene expression and metabolite concentration across different parts of *I. asarifolia* are particularly noteworthy.

## ACKNOWLEDGEMENTS

We thank the members of the plant taxonomy-anatomy lab who contributed to plant collection, and the molecular and allergy lab for their valuable support in the molecular work. We extend our appreciation to Dr. Tomoki Sando for sharing valuable information regarding the location of our plant study population. We thank Prof. Dr. Tokuko Haraguchi, Dr. Anthony E. Aiwonegbe, Dr. Rick Miller and two anonymous reviewers for their helpful comments on an earlier version of this manuscript.

### Funding

This research was funded by the Development and Promotion of Science and Technology Talents Project (scholarship awarded to Yanisa Olaranont), and from Mahidol University (MU's Strategic Research Fund: 2023, awarded to Paweena Traiperm and Alyssa B. Stewart). The funders had no role in study design, data collection and analysis, decision to publish, or preparation of the manuscript.

### Grant Disclosures

The following grant information was disclosed by the authors:
Development and Promotion of Science and Technology Talents Project.
Mahidol University, MU's Strategic Research Fund: 2023.

### Competing Interests

The authors declare that they have no competing interests.

### Author Contributions

- Yanisa Olaranont conceived and designed the experiments, performed the experiments, analyzed the data, prepared figures and/or tables, authored or reviewed drafts of the article, and approved the final draft.
- Alyssa B. Stewart conceived and designed the experiments, analyzed the data, authored or reviewed drafts of the article, and approved the final draft.
- Wisuwat Songnuan conceived and designed the experiments, analyzed the data, authored or reviewed drafts of the article, and approved the final draft.
- Paweena Traiperm conceived and designed the experiments, analyzed the data, authored or reviewed drafts of the article, and approved the final draft.

### Field Study Permissions

The following information was supplied relating to field study approvals (*i.e.*, approving body and any reference numbers):

Permission to collect plant specimens was granted by private land owners (Mr. Sorraton Sookmark).

## DNA Deposition

The following information was supplied regarding the deposition of DNA sequences:

The sequences are available at GenBank: PV410212–PV410219.

## Data Availability

The data is available in the Supplemental Files.

## Supplemental Information

Supplemental information for this article can be found online at http://dx.doi.org/10.7717/peerj.19692#supplemental-information.

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
