# Peer review of "Analysis of ergot alkaloid gene expression and ergine levels in different parts of Ipomoea asarifolia"

_PeerJ, doi:10.7717/peerj.19692_

## Round 0.1 · original submission · Major Revisions

All reviewers acknowledge the importance of this paper. However, they also point out some problems. I think that each reviewer's comments are reasonable and important for improving the paper. In particular, as reviewer 1 pointed out, I think you should add the controls in qPCR experiments. Please revise according to the reviewers' comments.

Reviewer 1 ·

Basic reporting

The article is well written and easy to read. The purpose is clearly stated and the results are clearly presented. Description of some important controls is missing from the qPCR section. Those controls and their significance will be discussed below.

Experimental design

The work is original and addresses an interesting question. My primary concern with the study pertains to controls in the qPCR experiment.
1. The qPCR results are novel and provide important information. Difficulty in interpreting significance of the results, however, arises in samples in which dmaW has relatively low expression levels, such as roots and stems. The difficulty arises because the qPCR section makes no mention of two important controls: a) no template control wells and b) sequencing of amplicons. In the absence of descriptions of these controls, the reader has to be skeptical of the data indicating very low concentrations of qPCR products. Products arising in latter rounds of qPCR often can be observed in no template control wells. (The authors also don’t report how many cycles they ran in each qPCR run.) Comparisons of curves generated in treatment wells with those generated in no template control wells is critical to determining the authenticity of the products. Sanger sequencing of amplicons also should be done to demonstrate the products are derived from the target locus and not from other potentially contaminating DNA.

Validity of the findings

I won't repeat my criticism of the qPCR data, but the lack of controls (or perhaps failure to include their description) does significantly affect the validity of the results.

A couple small points that would improve the article:

In figure 2 writing out the tissue names on the x axis would be much better than the abbreviations presently provided, especially since stem and seed both are abbreviated as S.

Since figure 4 is intended to test for a correlation, the inclusion of a line on the graph would strengthen the figure. The line would be helpful even in the observed absence of correlation.

Additional comments

I do not see the higher expression of dmaW in younger leaves (compared to older leaves) and the greater accumulation of ergine in older leaves (compared to younger) to be conflicting. Accumulation in the key word here. Ergine is a stable compound, and the older leaves have simply had longer to accumulate it. With respect to the observations in seeds, the authors hypothesis about environmental conditions in young seeds compared to older seeds seems very reasonable. Gene expression and alkaloid accumulation happen early in seed development.

·

Basic reporting

1.The "Abstract" shoud be restructured to conform with the standard format of the Journal by breaking it to the subsections:
Background
Methods
Results
Conclusion.
2. The arabic numerals in lines 21, 22, 26, 27 and 28 should be deleted and the respective sentences rewritten to allow for straight and continous reading.
3. References quoted between lines 43 and 52 should be updated with more recent ones.
4. Format "et al" in italics.
5. The sentences in lines 78 and 80 should be restructured to capture the scope of study without necessarily itemizing with numbers.
6. Recast the sentence in line 224 to give credits to the researchers oor authors rather than "the study"

7, References in lines 381-382 and lines 416-418 are incomplete.

Experimental design

The authors should consider reconstructing all the opening sentences in the "Methodology" section to 'reported speech' format.
Specifically, the word ’we’ should be deleted/replaced in the whole section to allow for variation in sentence construction in the varous paragraphs, eliminate redundancy and promote readability for international auduience.
For example,
line 86 should read: " Ipomoea asarifolia (Desr.) Roem. & Schult. was collected from......."

Line 135: Table 1 should be captioned:
"Some Examples of Genes and Primer Sequences"

Validity of the findings

no comments

Additional comments

I commend the authors for their extensive DNA analysis of the different plant parts of Ipomoea asarofolia

Reviewer 3 ·

Basic reporting

This manuscript examines a morning glory species for patterns of gene expression for ergot alkaloid synthesis by its fungal symbiont, in relation to the actual concentration of ergine. The results reveal a disconnect where tissues with lower levels of gene expression (mature leaves) have higher alkaloid concentrations and some tissues with high levels of expression have lower levels of actual alkaloids. The authors then discuss several hypotheses that might explain theses conflicting results.

This is a well-written and interesting paper, and follows from an earlier paper by the group on the distribution of the fungus across different host plant parts. Technically, I think that all of the approaches and analyses conducted are fine. I appreciated that in addition to dmaW gene expression, expression of several other fungal genes was also quantified. The data on both gene expression and ergine concentrations across a range of plant tissue types (not just seeds and leaves) are novel and interesting. The three possible hypotheses about why expression and alkaloid concentrations do not always match were interesting.

Experimental design

A second issue is that ergine is just one of many ergot alkaloids produced by Periglandula fungi, and the authors review previous reports of a wider range of ergot alkaloids in I. asarifolia. Only ergine is reported here and dmaW expression is not specific to ergine. The question then arises whether other ergot alkaloids show similar or divergent patterns of gene expression and alkaloid concentration? This seems to be a major gap – why were not other ergot alkaloids quantified? On line 316-317, it says that they were unable to obtain other types of ergot alkaloids. This may be due to local restrictions or costs of obtaining pure standards, but readers do not know if ergine concentrations are exceptional or reflective of the other ergot alkaloids. Also, given that there is a biochemical pathway for ergot alkaloid production, it is not clear if ergine is the end product or an intermediate step in the pathway, or even a breakdown product from other alkaloids. The interpretation of the results would be much clearer if additional ergot alkaloids were also quantified, even if it was only ergine vs. all other ergot alkaloids combined.

Line 232 discusses fungal biomass but I did not see anywhere how that was measured or if it was? The authors also mention fungal densities without any explanation of methodology. The correlation of gene expression or alkaloid concentrations with local fungal biomass is interesting, but it was not clear if the authors have collected quantitative data on density and biomass, or whether they are referring to anecdotal observations. Further, lines 230-233 suggest that expression is correlated with fungal biomass but I suspect that variable gene expression could reflect a wider range of conditions than just biomass. See 244.

Validity of the findings

I found that there were several areas that need to be clarified or reconsidered. The validity of the findings is based in part on my comments above about measuring only ergine and not other alkaloids. I understand that there could be legal restrictions or financial or technical reasons that a wider range of alkaloids were not quantified, but that lack nevertheless compromises the interpretation of the ergine results.

The discussion of fungal biomass and density was unclear if there was quantitative data or just observation.

Additional comments

One point is that this paper is very morning glory/Periglandula-centric and could be pitched more to researchers on different systems where patterns of gene expression and concentration of products within host plants do not strongly correlate. Why should someone not interested in Ipomoea and Periglandula read this paper?

·

Basic reporting

Review of “Analysis of ergot alkaloid gene expression and ergine levels in different parts of Ipomoea asarifolia” by Yanisa Olaranont, Alyssa B. Stewart, Wisuwat Songnuan, and Paweena Traiperm.

The research conducted by Olaranont and colleagues addresses an important knowledge gap in our understanding of the morning glory – clavicipitaceous fungi symbiosis. For decades, it was thought morning glories themselves produced ergot alkaloids, leading to extensive surveys that tested seeds from numerous morning glory species. In most morning glory species (e.g. Ipomoea tricolor), these fungi are cryptic. However, recent studies have identified fungal colonies in the Clavicipitaceae on the adaxial side of leaves, where they are associated with secretory glands and produce ergot alkaloids using plant metabolites. This finding aligns with our understanding of similar grass-fungus interactions. While the grass – fungus symbiosis has been extensively studied, the morning glory – fungal relationship remains relatively unexplored, which makes this study a valuable contribution to the field.

In conjunction with the companion paper by Olarnont et al. (2022), this work significantly advances our understanding of the Ipomoea – Periglandula symbiosis by carefully examining all plant parts in freshly collected tissues of native Ipomoea asarifolia. This research documents the location of fungal mycelia with hallmarks of Periglandula using three histological methods, employs PCR to detect the important gene dmaW (a key gene in the ergot alkaloid biosynthetic pathway). The present study builds on this by using quantitative PCR to assess dmaW expression levels while also analyzing ergine concentration across plant tissues. This study reveals fascinating aspects of the fungal behavior within host plants, including variable gene expression, ergine concentration targeting specific tissues, and low levels of fungal mycelia across the plants that could facilitate vertical transmission, ensuring continuity of the fungi across plant generations.

Previous studies focused on the presence of ergot alkaloid in mature seeds, sometimes even from herbarium specimens and assays of dmaW were limited. The comprehensive approach taken here – quantifying dmaW expression and ergine concentrations in fresh plant material -- offers significant insights into which tissues are producing this ergot alkaloid and how this relates to gene expression, clarifying the fungal role the symbiosis.

This research presented is exceptional in multiple aspects. The research objectives are clear, and the methods employed were highly rigorous, with the authors overcoming technical challenges such as isolating fungal RNA signals within plant material and developing precise HPLC systems for detecting a key ergot alkaloid. The manuscript is well-written, and after thorough review, I found no need for textual edits for clarifications. The figures are clear and well-explained, with supplementary figures effectively supporting the primary results. This work represents a high-quality contribution to our understanding of the morning glory – clavicipitaceous fungi symbiosis.

Minor Concerns

1. While I have not personally conducted this type of work, precautions are generally necessary to prevent contamination during tissue grinding for ergot alkaloid quantification. It would be beneficial for the authors to clarify how contamination was managed.

2. In the discussion, it would strengthen the manuscript to provide additional ecological details regarding herbivory and seed predation in morning glories. For instance, bruchid beetle larvae are known seed predators of Ipomoea pes-caprae, as species closely related to the study organism and often the focus of many relevant studies. Since elevated ergine levels were observed in young leaves and seeds – possibly an adaptive response to herbivory – a discussion of known herbivors and seed predators of morning glories would add valuable context.

Larger Context

Although this may reflect my perspective, developing the evolutionary and phylogenetic context for the relationship between Ipomoea asarifolia and Periglandula could underscore the importance of this research. Specifically, a stronger connection between the study’s significant and the presentation in Beaulieu et al. (2021; Diversification of ergot alkaloids and heritable fungal symbionts in morning glories) and related studies could enhance our understanding of this symbiosis. We know that species of Convolvulaceae that participate in the ergot alkaloid producing symbiosis (ergot positive species) are restricted to members of the tribe Ipomoeeae. Across the tribe, ergot-positive species form clades where ergot alkaloid production is a characteristic trait. This includes a large clade dominated by Argyreia species (well-represented in Thailand). Ipomoea asarifolia is a member of another large clade of ergot-positive species, known informally as the ‘Pes-caprae’ group. Evidence suggests the symbiosis arose in the common ancestor of these ergot positive clades. Furthermore, alkaloid profiles differ between clades, indicating coevolution between fungi and plants. Although Beaulieu et al. (2021) did not include an alkaloid profile for Ipomoea asarifolia, it is known that ergobalansine and ergine are dominant alkaloids produced by closely related ‘Pes caprae’ species. Beaulieu et al. (2021) also proposed a hypothesis regarding the positive correlation between seed size and ergot alkaloid production. While I. asarifolia seed size was not presented, the seeds of the study organism are similar in size to the Ipomoea pes-caprae.

Overall, the authors have developed an outstanding manuscript that is rooted in rigorous research. I am confident that Olaranont and colleagues will know best how to use these suggestions to broaden the evolutionary context of this contribution, drawing on our knowledge of morning glory phylogeny and the potential coevolution of this fascinating ergot alkaloid-producing symbiosis.

Experimental design

see above

Validity of the findings

see above

---

## Round 0.2 · Major Revisions

It seems that the track change does not adequately indicate the location of the modifications. Please clarify them on the tracked changes compared to the initial manuscript. Most importantly, you must provide convincing experimental evidence for the identity of the root and stem products, especially by providing DNA sequences of damW amplicons from roots and stems.

Reviewer 1 ·

Basic reporting

The tracked changes document shows only one tracked change in the text, whereas the authors claimed to have made several substantial changes.

Experimental design

I'm providing only brief comments on the revised manuscript only in some of the boxes.

Validity of the findings

Contrary to the connotation of the very general text of their rebuttal letter and the revised methods, the authors have not provided DNA sequence for dmaW amplicons produced from roots and stems. They provided sequence only for young leaf and young seed which were never in question based on the amount of amplification from those tissue and also previous work on this symbiosis. Without compelling experimental evidence for the identity of the product in roots and stems, the authors should be much more conservative with their interpretation of their qPCR results.

On line 135, the authors claim to have “BLAST searched each primer against the whole genome of I. asarifolia to confirm that they are specific to the fungus, with no matches in the plant genome.“ The authors should provide an accession number and citation for the I. asarifolia genome. I cannot find a publicly available genome sequence for I. asarifolia or even a single reference to it.

·

Basic reporting

pass

Experimental design

pass

Validity of the findings

pass

---

## Round 0.3 · Minor Revisions

We have confirmed that you have provided DNA sequence data. We have received minor comments from Reviewer 1, which are also necessary to improve your manuscript. Please correct them and resubmit your manuscript.


· · Academic Editor

Reject

The DNA sequence data (accession numbers) used for the analysis are not provided. Without the DNA sequence data, it is difficult to guarantee the authenticity of the paper, and therefore it cannot be accepted by PeerJ.

Reviewer 1 ·

Basic reporting

The statement about blasting primers and not finding a match to plant sequences is still misleading, because there are no plant sequences available to which the primers could match. Perhaps reword it to say "...with no match to the plant sequences from several related plant species, providing a limited level of support for their specificity to the fungus."

Experimental design

n/a

Validity of the findings

n/a

Additional comments

n/a

---

## Round 0.4 · accepted · Accept

I have confirmed that the authors have addressed all of the reviewers' concerns. Therefore, this manuscript is ready for publication.

Reviewer 1 ·

Basic reporting

n/a

Experimental design

n/a

Validity of the findings

n/a

Additional comments

I reviewed the changes and have no new criticism of the manuscript.